spectroscopy

spectrofluorometric, olsalazine, sulfasalazine, quenching

**Author for correspondence:**
Heba Elmansi
e-mail: dr_heba85@hotmail.com

# Studying the quenching resulted from the formation of an association complex between olsalazine or sulfasalazine with acriflavine

## Manar Tolba and Heba Elmansi

Department of Pharmaceutical Analytical Chemistry, Faculty of Pharmacy, Mansoura University, Mansoura 35516, Egypt

(iD) HE, 0000-0002-3953-7169

We report the detection and quantification of important ulcerative colitis drugs olsalazine (OLS) and sulfasalazine (SUL) by the spectrofluorometric method. The proposed method was optimized and validated by using the quenching effect on the acriflavine fluorescence. The method was applied on the detection and quantification of OLS and SUL under optimized conditions showing the calibration curves were linear (range: 1.0–10.0 µg ml$^{-1}$), with correlation coefficients $R^2$ of 0.9999 for both drugs. The limits of detection (LOD) and quantification (LOQ) were 53 and 104 ng ml$^{-1}$ for the OLS and 160 and 315 ng ml$^{-1}$ for the SUL. This method permitted the analysis of OLS and SUL in their pure and pharmaceutical forms. The proposed spectrofluorimetric method was also evaluated against 'green' criteria and all the experimental results make it an eco-friendly and safe method for the detection of OLS and SUL.

# 1. Introduction

Olsalazine (OLS) and sulfasalazine (SUL) (figure 1) are effective agents used for treating ulcerative colitis either alone or with corticosteroids and for maintaining remission [1]. OLS is disodium 3,3′-diazenediylbis(6-hydroxybenzoate) [2] and is listed in both the British Pharmacopoeia [2] and European Pharmacopoeia (Ph. Eur) [3]. OLS consists of two molecules of mesalazine linked with an azo bond, is activated in the colon to the active form of mesalazine, and is used as the sodium salt at an adult dose of 250–500 mg twice daily [1].

This article has been edited by the Royal Society of Chemistry, including the commissioning, peer review process and editorial aspects up to the point of acceptance.

**Figure 1.** The structural formulae for OLS, SUL and AC.

SUL is 2-hydroxy-5-[2-[4-(pyridin-2-ylsulphamoyl) phenyl] diazenyl] benzoic acid [2] and is also listed in the Ph. Eur [3] and US Pharmacopoeia [4]. SUL consists of a sulfonamide, sulfapyridine and 5-aminosalicylic acid (mesalazine), and its activity is considered to mainly involve the 5-aminosalicylic acid moiety that is released in the colon by bacterial metabolism, although intact SUL has some anti-inflammatory properties [1]. SUL is also used as a disease-modifying drug in the treatment of severe or progressive rheumatoid arthritis. In inflammatory bowel disease, the usual initial adult dose of sulfasalazine is 1–2 g orally four times daily [1].

Acriflavine (AC) is 3,6-diamino-10-methylacridin-10-ium chloride [5], which is a fluorescent dye used as a local antiseptic, antibacterial drug and anti-cancer agent as well as a fluorescent and absorptive reagent and in preparation of dyes [6,7]. Acriflavine has been used for the quantitative assessment of vitamin C and furosemide [8,9].

The investigated drugs have been evaluated in the literature by using spectrophotometric [10–13], spectrofluorimetric [14,15], HPLC [16–18] and electrochemical methods [19–21]. HPLC requires expensive instruments and large volumes of hazardous solvents that require problematic disposal. Moreover, time-consuming or tedious sample pre-treatment steps may be required. Spectrophotometric methods suffer from low sensitivity or a narrow range. Additionally, some methods are indirect and involve multi-reagent reactions or heating, which may affect the accuracy and precision. Electrochemical methods are complicated and require experienced operators. Alternatively, spectrofluorimetry is a

convenient technique in the pharmaceutical analysis due to its simplicity, low cost and wide availability [22]. Fluorescence quenching has recently received interest for use in biochemical applications because the decrease in fluorescence of a compound is possibly quantitatively related to its concentration [23].

The study aim was to investigate a proposed method based on acriflavine quenching for rapid quantification of OLS and SUL. To date, no spectrofluorimetric methodology for OLS has been reported. A further goal of this study was to evaluate the method for its potential harm to the environment. Most recently developed chemical and analytical procedures consider the 'green' concept and strive to reduce sample preparation, solvent consumption, energy usage and waste generation [24]. Accordingly, we evaluated the proposed method against green criteria (i.e. environmentally responsible) by using three evaluation tools: the national environmental method index (NEMI) [25], analytical eco-scale [26] and green analytical procedure index (GAPI) [27].

# 2. Materials, reagents and methods

## 2.1. Materials, reagents and standard solutions

An aqueous solution of acriflavine ($8 \times 10^{-4}$ mol l$^{-1}$; Sigma-Aldrich, Germany) consisting of 0.0208 g acriflavine in 100 ml distilled water was diluted to a concentration of $8 \times 10^{-6}$ mol l$^{-1}$. Acetic acid, boric acid, phosphoric acid and sodium hydroxide were purchased from (El-Nasr Pharmaceutical Chemicals Co., Abu-Zabaal, Cairo, Egypt). Sodium dodecyl sulfate (SDS; 95%), tween-80 and β-cyclodextrin (β-CD) were purchased from El-Nasr Pharmaceutical Chemicals Company (ADWIC) (Abu Zaabal, Egypt) while cetrimide (CTAB; 99%) were purchased from Winlab (UK). Britton–Robinson buffer (BRB) solutions were prepared for use in the pH range from 2.0 to 12.0, and pH adjustments were made by using a pH-meter. OLS and SUL with purities of 100.22% and 99.82% were purchased from Sigma-Aldrich, Merck KGaA, Darmstadt, Germany. The solvents used were of analytical grade.

Dipentum: 250 mg OLS/capsule (UCB, Inc., Rochester, NY, USA) was used. For Alaven Pharmaceutical LLC, Marietta USA. Colosalazine-EC: 500 mg SUL/capsule was manufactured by Arab Caps for ACDIMA International Trading, El Mohandseen, Cairo, Egypt.

The standard solutions of both OLS and SUL were prepared by dissolving 10.0 mg of the drug in distilled water or methanol, respectively, in a 100 ml volumetric flask and then diluted to volume with the respective solvent. Further dilutions were carried out to reach the studied concentration range.

## 2.2. Equipment

— Spectrofluorimetric measurements were performed by using a Cary Eclipse spectrofluorimeter equipped with a Xenon-arc lamp adjusted to a medium voltage (700 V). The slit width of the monochromator was set to 10 nm, and a 1 cm quartz cell was used.
— pH-meter, model AD11P (Adwa, Romania).
— Bath sonicator (SONICOR SC-101TH).

## 2.3. Analytical procedures

### 2.3.1. Calibration-graph construction

To construct the graphs for OLS and SUL, 1.0 ml BRB solution at pH 7 and 1.1 ml acriflavine ($8 \times 10^{-6}$ mol l$^{-1}$) were added to standard-solution aliquots of both drugs in 10 ml volumetric flasks and diluted to reach final drug concentrations ranging from 1.0 to 10.0 µg ml$^{-1}$. The quenching of acriflavine fluorescence was measured at 504 nm after excitation at 260 nm. Relative fluorescence intensity (RFI) was plotted versus the final drug concentration, and regression equations were derived. The sensitivity of the instrument was adjusted to a medium voltage (700 V).

### 2.3.2. Determination of OLS and SUL in pharmaceuticals

Ten Dipentum 250-mg capsules were evacuated accurately and then ground. An amount of the powder equivalent to 50 mg OLS was weighed into a 100 ml volumetric flask, sonicated for 30 min in 50 ml of water and then filled to volume with water. The sample solution was filtered, and then the filtrate was diluted and analysed per the previous procedure for calibration-graph construction.

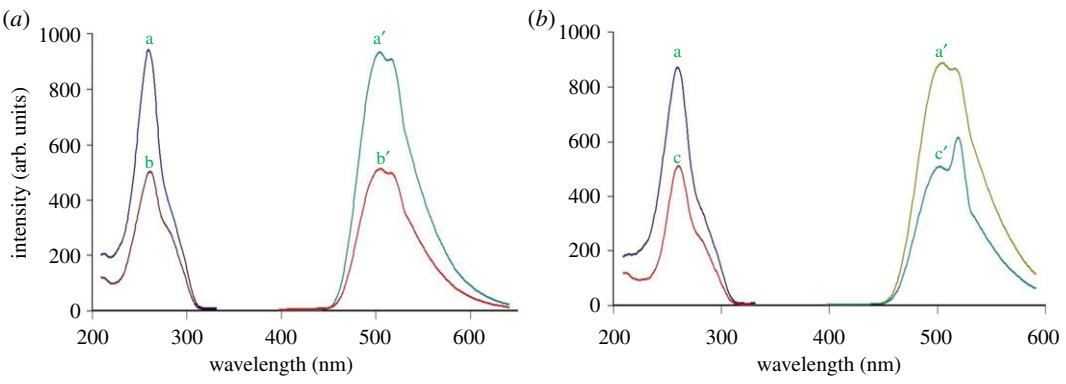

**Figure 2.** Excitation and emission fluorescence spectra of (*a*) a, a′ AC (8.8 × 10$^{-7}$ M) b, b′ OLS (10.0 µg ml$^{-1}$). (*b*) a, a′ AC (8.8 × 10$^{-7}$ M) c, c′ SUL (10.0 µg ml$^{-1}$).

Ten Colosalazine-EC tablets were finely powdered. Then, 50 mg of SUL was transferred to a 100 ml volumetric flask, and 50 ml methanol was added. The solution was sonicated for 30 min and then filled to volume with methanol. The solution was filtered and diluted quantitatively with the same solvent to the final concentration and treated as in the previous procedure for calibration-graph construction.

# 3. Results and discussion

Acriflavine, as a fluorescent dye, has been used as a reagent for spectrofluorimetric quantitative estimation of OLS and SUL. The dye has a high inner fluorescence intensity at 504 nm after excitation at 260 nm. The proposed method relies on quenching of the native acriflavine fluorescence by OLS and SUL in the buffered medium. Figure 2 shows the excitation and emission fluorescence spectra of acriflavine before and after the reactions with OLS and SUL.

## 3.1. Optimization of experiments

The factors that may affect the formed-ion-associated complex have been studied and optimized to reach the maximum sensitivity and robustness of the proposed method. These variables include pH and volume of buffer, reagent volume, diluting solvent and surfactant impact.

### 3.1.1. Buffer pH and volume

Solution pH is considered a critical parameter of the medium and is required to permit protonation of acriflavine and the reaction to proceed [8,9]. BRB has been investigated over the pH range of 3–10, and it was found that the optimum pH was 7 for both OLS and SUL (figure 3*a*). Moreover, different volumes of buffers were examined to yield the maximum RFI and optimal spectral shape. Volumes from 0.5 ml up to 2.5 ml were studied, and no significant effect from increasing the volume of buffer was observed since the reading remained constant, so 1 ml of BRB pH 7 was added for both drugs.

### 3.1.2. Reagent volume

Increasing the volumes of acriflavine solution (8 × 10$^{-6}$ M) up to 1.0 ml increased the fluorescence quenching. Maximum values were obtained when using volumes equal to 1.1 ml for both drugs (figure 3*b*).

### 3.1.3. Diluting solvents

The effects of different solvents on the ion-associated complex were investigated. Water, methanol, ethanol and acetonitrile were examined. Methanol as the diluting solvent gave the maximum RFI, and other solvents resulted in lower quenching readings and turbid solutions. However, water was the solvent of choice because of its availability, low cost and eco-friendly properties.

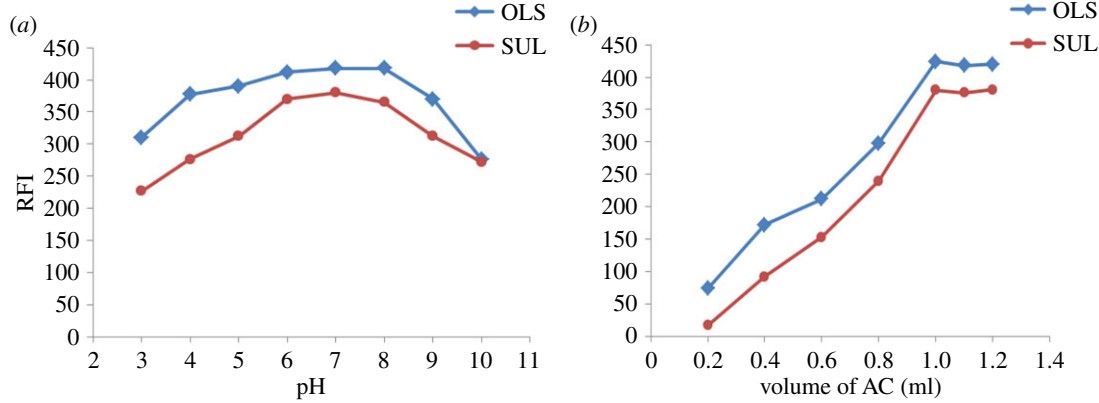

**Figure 3.** Effect of (a) pH of BRB and (b) volume of AC ($8 \times 10^{-6}$ M) on the fluorescence quenching by OLS (10 µg ml$^{-1}$) and SUL (10 µg ml$^{-1}$).

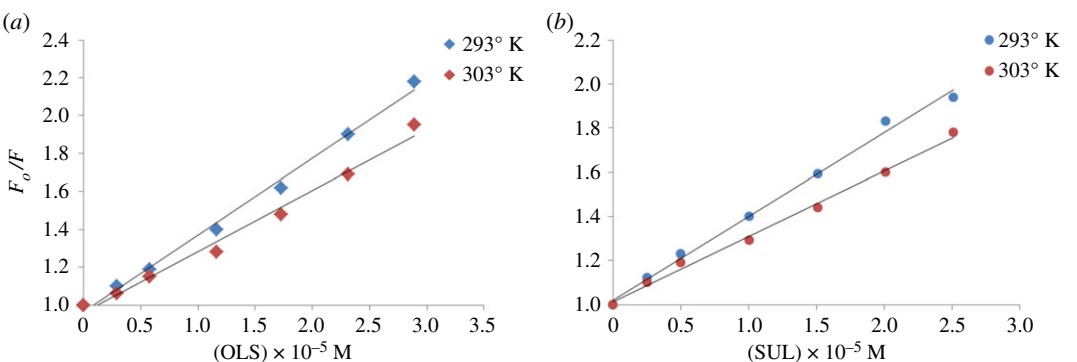

**Figure 4.** (a,b) Stern–Volmer plots of the studied drugs at two different temperature settings.

### 3.1.4. Surfactant impact

All the studied surfactants except SDS did not significantly affect the RFI of the ion-associated complex of OLS and SUL with acriflavine. Although SDS increased the RFI of the ion-associated complex, it could not be used as it resulted in non-reproducible results. Accordingly, no surfactant was included in the method.

## 3.2. Determination of quenching mechanism and stoichiometry of the reaction

Fluorescence quenching resulting from the complex formation may be caused by different molecular interactions, including excited-state reactions, molecular rearrangements, ground state complex formation and collisional quenching [23]. To assess the type of quenching that occurred, Stern–Volmer plots were constructed according to the following equation:

$$\frac{F_o}{F} = 1 + K_{SV}[C],$$

where $F$ and $F_o$ are the relative fluorescence intensities of acriflavine with and without the drug, respectively, $K_{SV}$ is the Stern–Volmer constant and [C] is the molar concentration of the drug.

The Stern–Volmer plots were graphed by plotting $F_o/F$ against [C], and the plots were linear ($r = 0.990$–$0.997$) (figure 4), which predicts the occurrence of dynamic or static quenching. Additionally, the temperature dependency was investigated by constructing Stern–Volmer plots at two different temperatures (figure 4). Table 1 shows a decrease in $K_{SV}$ with increasing temperature, which is indicative of static quenching [23]. We concluded that non-fluorescent complexes are formed between acriflavine and OLS or SUL. These complexes have decreased stability at higher temperatures. Furthermore, the bimolecular quenching constants ($K_q$) were calculated to indicate the

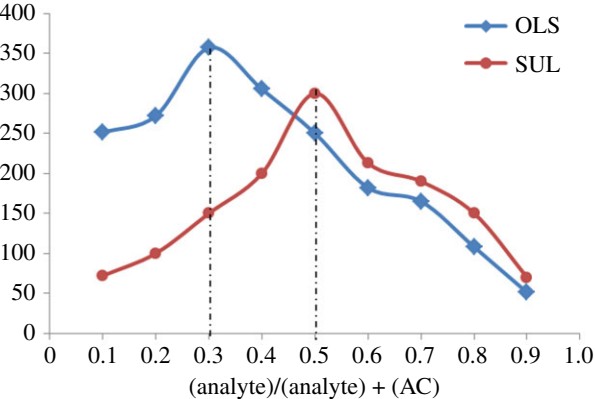

**Figure 5.** Continuous variation plots for determination of stoichiometry of reaction of the studied drugs and AC ($8 \times 10^{-6}$ M).

**Table 1.** Stern–Volmer parameters for the reaction between the studied drugs and acriflavine.

| drug | temperature (°K) | Stern–Volmer quenching constant $K_{sv} \times 10^5$ (l mol$^{-1}$) | correlation coefficient ($r$) | bimolecular quenching constant $k_q \times 10^{13}$ (l mol$^{-1}$ s$^{-1}$) |
|---|---|---|---|---|
| OLS | 293 | 0.41 | 0.997 | 0.81 |
| | 303 | 0.32 | 0.990 | 0.63 |
| SUL | 293 | 0.38 | 0.997 | 0.76 |
| | 303 | 0.30 | 0.997 | 0.59 |

fluorescence efficiency according to the following equation [28]:

$$K_q = \frac{K_{SV}}{t_0},$$

where the fluorescence lifetime of acriflavine is $5 \times 10^{-9}$ s, so $K_q$ was found to range from 0.59 to 0.81 $\times 10^{13}$ l mol$^{-1}$ s$^{-1}$.

This value is greater than $1 \times 10^{10}$, which showed that static quenching occurred through molecular binding and complex formation.

To determine the stoichiometric ratio between the studied drugs and acriflavine, Job's method of continuous variation was used [29]. Equimolar solutions ($8 \times 10^{-6}$ M) were prepared from the drug and acriflavine. The solutions were mixed in different molar ratios with a constant total molar concentration. We found that the ratio of acriflavine to the drug was 1 : 2 and 1 : 1 for OLS and SUL, respectively, as shown in figure 5. The ratio could be deduced by the presence of two carboxylic groups in OLS and only one carboxylic group in SUL. At the specified pH, the positively charged nitrogen atom in acriflavine reacts with the carboxylate anion of OLS or SUL. Hence, electrostatic forces contribute to ion-pair complex formation, as illustrated in scheme 1 [8,9].

## 3.3. Validation of the proposed method

International Conference on Harmonization (ICH) guidelines [30] were used to validate the method as follows.

A calibration curve between the RFI and the concentration in µg ml$^{-1}$ was constructed. As per the ICH guidelines [30], the correlation coefficient of the regression equation, intercept and slope of the regression line, linear range, the standard deviation of the residuals, limits of quantitation, limits of detection and the method standard deviation are presented in table 2. All of these parameters proved the linearity of the calibration graphs.

Accuracy was evaluated by comparing the analysis results for OLS and SUL in pure form with those obtained from previously reported methods [10,13]. Statistical analysis of the results showed no significant differences in accuracy and precision between the proposed method's results and the other methods' results as assessed by Student's $t$-test and the variance ratio test ($F$-test) [31] (table 3).

**Scheme 1.** Mechanism for the formation of an ion-associated complex between acriflavine and sulfasalazine.

**Table 2.** Analytical performance data for the proposed approach.

| parameters | OLS | SUL |
|---|---|---|
| linearity range ($\mu$g ml$^{-1}$) | 1.0–10.0 | 1.0–10.0 |
| intercept ($a$) | 30.060 | −9.800 |
| slope ($b$) | 38.666 | 39.800 |
| correlation coefficient ($r$) | 0.9999 | 0.9999 |
| s.d. of residuals ($S_{y/x}$) | 0.796 | 1.613 |
| s.d. of intercept ($S_a$) | 0.620 | 1.255 |
| s.d. of slope ($S_b$) | 0.102 | 0207 |
| percentage relative standard deviation, % RSD | 0.52 | 0.46 |
| percentage relative error, % error | 0.22 | 0.19 |
| limit of detection, LOD (ng ml$^{-1}$) | 53 | 104 |
| limit of quantitation, LOQ (ng ml$^{-1}$) | 160 | 315 |

The comparison method for OLS is based on spectrophotometric measurements in 0.1 N NaOH to give a yellowish-orange colour [10]. SUL was measured in 1 : 1 water : methanol at 359 nm [13].

Inter-day and intra-day precision data for assessment of OLS and SUL by the proposed spectrofluorimetric method are presented in table 4 and show low percentage relative standard deviations and percentage relative errors, which are indicative of a precision method.

**Table 3.** Application of the proposed approach for the assessment of OLS and SUL in pure forms.

| sample | amount taken (μg ml$^{-1}$) | amount found (μg ml$^{-1}$) | % found | comparison methods [10,13] |
|---|---|---|---|---|
| OLS | 1.0 | 1.007 | 100.71 | 99.63 |
| | 2.0 | 2.016 | 100.79 | 99.55 |
| | 4.0 | 3.981 | 99.53 | 101.20 |
| | 6.0 | 5.973 | 99.55 | |
| | 8.0 | 8.016 | 100.20 | |
| | 10.0 | 10.007 | 100.07 | |
| x̄ ± s.d. | | | 100.14 ± 0.54 | 100.13 ± 0.93 |
| *t* | | | 0.03 | |
| *F* | | | 2.93 | |
| SUL | 1.0 | 1.000 | 100.00 | 100.32 |
| | 2.0 | 2.005 | 100.25 | 98.78 |
| | 4.0 | 4.015 | 100.38 | 99.01 |
| | 6.0 | 6.000 | 100.00 | |
| | 8.0 | 7.935 | 99.18 | |
| | 10.0 | 10.045 | 100.45 | |
| x̄ ± s.d. | | | 100.47 ± 0.42 | 99.37 ± 0.83 |
| *t*\* | | | 1.61 | |
| *F*\* | | | 3.22 | |

N.B. Each result is the average of three separate determinations.
\*The tabulated *t* and *F* values are 2.36 and 5.79, respectively, at *p* = 0.05 [31].

**Table 4.** Precision data for the assessment of the studied drugs by the proposed method.

| amount taken (μg ml$^{-1}$) | % found | % RSD | % error |
|---|---|---|---|
| OLS | | | |
| intra-day; 2.0 | 99.23 ± 0.36 | 0.36 | 0.21 |
| 4.0 | 100.11 ± 0.51 | 0.51 | 0.29 |
| 6.0 | 99.75 ± 0.23 | 0.23 | 0.13 |
| inter-day; 2.0 | 100.35 ± 0.96 | 0.96 | 0.55 |
| 4.0 | 99.77 ± 1.02 | 1.02 | 0.59 |
| 6.0 | 101.35 ± 1.12 | 1.11 | 0.64 |
| SUL | | | |
| intra-day; 4.0 | 99.65 ± 0.45 | 0.45 | 0.26 |
| 6.0 | 100.42 ± 0.62 | 0.62 | 0.36 |
| 8.0 | 99.66 ± 0.80 | 0.80 | 0.46 |
| inter-day; 4.0 | 98.54 ± 1.10 | 1.12 | 0.64 |
| 6.0 | 99.63 ± 0.82 | 0.82 | 0.48 |
| 8.0 | 100.98 ± 1.08 | 1.07 | 0.62 |

The method was also used to analyse the studied drugs in their pharmaceutical preparations, and no interference from common excipients was observed, which supports good method selectivity. Moreover, the studied preparations were analysed by other methods for comparison [10,13], and no significant

**Table 5.** Application of the proposed approach for the assessment of the pharmaceutical dosage forms.

| sample | amount taken ($\mu g\ ml^{-1}$) | amount found ($\mu g\ ml^{-1}$) | % found | comparison methods [10,13] % found |
|---|---|---|---|---|
| Dipentum 250 mg capsules | 2.0 | 2.030 | 101.50 | 101.00 |
| (250 mg OLS/capsule) | 4.0 | 3.996 | 99.90 | 100.55 |
| | 6.0 | 6.020 | 100.33 | 99.11 |
| $\bar{x} \pm$ s.d. | | | $100.58 \pm 0.83$ | $100.22 \pm 0.99$ |
| $t$ | | | 0.48 | |
| $F$ | | | 1.42 | |
| nominal content | | | 251.45 | |
| Colosalazine-EC tab. | 4.0 | 4.030 | 100.75 | 100.33 |
| (500 mg SUL/tablet) | 6.0 | 6.040 | 100.67 | 99.35 |
| | 8.0 | 7.998 | 99.98 | 100.12 |
| $\bar{x} \pm$ s.d. | | | $100.47 \pm 0.42$ | $99.93 \pm 0.52$ |
| $t^*$ | | | 1.38 | |
| $F^*$ | | | 1.49 | |
| nominal content | | | 502.35 | |

N.B. Each result is the average of three separate determinations.

*The tabulated $t$ and $F$ values are 2.78 and 19, respectively, at $p = 0.05$ [31].

differences in accuracy and precision between the methods by Student's $t$-test and the variance ratio test ($F$-test) were found [31].

## 3.4. Application to pharmaceutical preparations of OLS and SUL

The proposed spectrofluorimetric method was applied successfully to the assay of OLS and SUL in their pure forms. Accordingly, Dipentum 250 mg capsules and Colosalazine-EC tablets were analysed to assess the selectivity of the method and lack of interference from additives and excipients. The mean content was $100.58 \pm 0.83\%$ and $100.47 \pm 0.42\%$ for the Dipentum capsules and Colosalazine-EC tablets, respectively (table 5). The additives included in these dosage forms were magnesium stearate, colloidal silicon dioxide, polyvidone 30, crospovidone and 99.5% ethanol.

## 3.5. Evaluation of the eco-friendly properties of the proposed method

### 3.5.1. Green analytical procedure index

The GAPI tool uses new criteria regarding safety and health that can be considered when developing analytical methods [27] and uses five pentagrams to quantify the environmental impact of each step of an analytical method with a colour code. Interpretation of the GAPI pentagrams for the proposed method revealed that most fields were coloured green (low impact) and only two were yellow (medium impact) and two were red (high impact) (table 6).

### 3.5.2. National environmental method index

Assessment by this tool relies on the evaluation of the four criteria for the chemicals used in a method (table 6). The solvent used in the proposed method is water, which is obviously an eco-friendly solvent as stated by the US Environmental Protection Agency's Toxic Release Inventory (Emergency Planning and Community Right-to-Know Act, 2004; Code of Federal Regulations, 2014). It is not listed as a persistent bioaccumulative toxic or hazardous solvent, and the pH used during the study is 7, which is not corrosive. Moreover, the amount of waste generated from the two methods was less than 50 g per sample. Hence, the proposed overall method fulfilled the goal of four green quadrants of the greenness profile.

**Table 6.** Results for evaluation of greenness of the proposed approaches.

| **1. Green analytical procedure index (GAPI) [27]** |
|---|

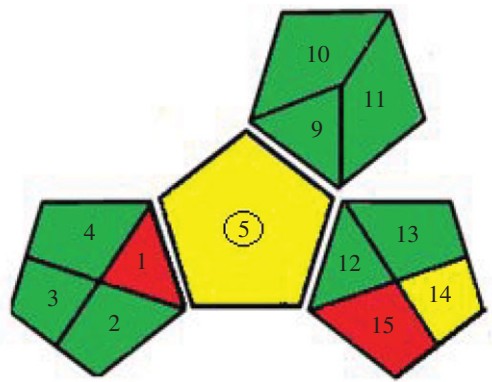

| **2. NEMI pictogram [25]** |
|---|

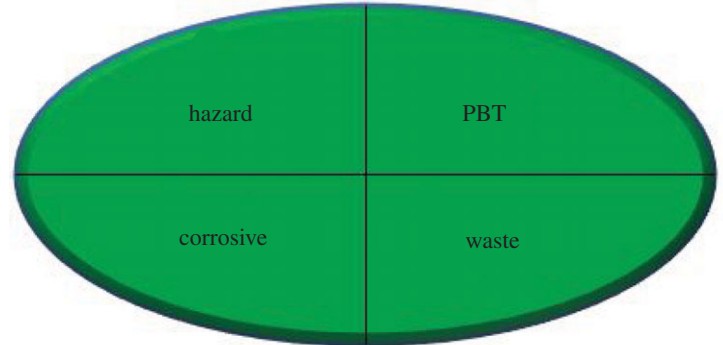

| **3. Analytical eco-scale score [26]** | | | |
|---|---|---|---|
| item | no. of pictogram | word sign | penalty points |
| reagent; volume (ml) | | | |
| AC; 1.1 ml | 3 | danger | 6 |
| BRP; 1 ml | 1 | danger | 2 |
| spectrofluorimeter | | | 0 |
| occupational hazard | | | 0 |
| waste | | | 3 |
| total penalty points | | | 11 |
| analytical eco-scale score | | | 89 |

### 3.5.3. Analytical eco-scale

The eco-scale method assigns penalty points for each step [26,32]. The total number of penalty points is determined by multiplying the sub-total penalty points for a given amount and hazard (table 6). The score of the proposed spectrofluorimetric method was 89, which indicated an excellent green method.

## 4. Conclusion

The proposed spectrofluorimetric method was effectively used to estimate OLS and SUL by measuring their quenching effect on the acriflavine fluorescence. The proposed method was optimized, validated and applied successfully for the pharmaceutical preparations of the OLS and SUL. The static quenching mechanism was confirmed by the decrease in Stern–Volmer constants with the increasing temperature. The method is facile, sensitive and economical alternative to previously reported

analytical methods that consume expensive or hazardous solvents with complicated instrumentation. Additionally, the spectrofluorimetric method is 'green' thereby recommended for safety, applicability and eco-friendly nature.

Data accessibility. The study data were deposited in the Dryad Digital Repository [33]: Elmansi, Heba; Tolba, Manar (2021), Calibration curves sheets, Dryad, Dataset, https://doi.org/10.5061/dryad.v41ns1rv6.

Authors' contributions. M.T. performed the laboratory work and statistical calculations. H.E. drafted the manuscript. Both authors designed the study and revised and approved the manuscript.

Competing interests. We declare we have no competing interests.

Funding. No funding institutions supported this research.

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
