## [Peer Review File · Royal Society Open Science]

Review History

RSOS-210110.R0 (Original submission)

Review form: Reviewer 1

Is the manuscript scientifically sound in its present form?

Yes

Are the interpretations and conclusions justified by the results?

Yes

Is the language acceptable?

Yes

Do you have any ethical concerns with this paper?

No

Have you any concerns about statistical analyses in this paper?

No

Recommendation?

Accept with minor revision (please list in comments)

Comments to the Author(s)

I highly recommend the manuscript for publication after minor revision as below.

Suggestion to Authors

1- Please modify the Abstract section as below.

We report the detection and quantification of important ulcerative colitis drugs olsalazine (OLS) and sulfasalazine (SUL) by the Spectrofluorometric method. The proposed method was optimized and validated by using the quenching effect on the acriflavine fluorescence. The method was applied on the detection and quantification of OLS and SUL under optimized conditions showing the calibration curves were linear (Range: 1.0–10.0 µg/mL), with correlation coefficients $R^2 = 0.9999$ for both the drugs. The limits of detection (LOD) and quantification (LOQ) were 53 ng/mL & 104 ng/mL for the OLS and 160 ng/mL & 315 ng/mL for the SUL. This method permitted the analysis of OLS and SUL in their pure and pharmaceutical forms. The proposed spectrofluorimetric method was also evaluated against “green” criteria and all the experimental results make it an eco-friendly and safe method for the detection of OLS and SUL.

2- Please modify the Conclusion section as below.

The proposed spectrofluorimetric method was effectively utilized to estimate OLS and SUL by measuring their quenching effect on the acriflavine fluorescence. The proposed method was optimized, validated and applied successfully for the pharmaceutical preparations of the OLS and SUL. The static quenching mechanism was confirmed by the decrease in Stern–Volmer constants with the increasing temperature. The method is facile, sensitive and economical alternative to previously reported analytical methods that consume expensive or hazardous solvents with complicated instrumentation. Additionally, the spectrofluorimetric method is “green” thereby recommended for safety, applicability and eco-friendly nature.

Review form: Reviewer 2**Is the manuscript scientifically sound in its present form?**

Yes

Are the interpretations and conclusions justified by the results?

Yes

Is the language acceptable?

Yes

Do you have any ethical concerns with this paper?

No

Have you any concerns about statistical analyses in this paper?

No

Recommendation?

Accept with minor revision (please list in comments)

Comments to the Author(s)

It may be published after a revision considering the following points:

(1) The purity of olsalazine and sulphasalazine should be written under section 2.1

- (2) Under section 2.1 "OLS and SUL were kindly provided by sigma" should be changed to "OLS and SUL were purchased from Sigma"; in addition country should be added.
- (3) Under material, reagent and standard solution: "filling to volume" should be changed to "diluted to volume".
- (4) Effect of surfactant should be added under optimization of experiment section 3.1
- (5) The sensitivity of instrument should be added under section 2.3.1 calibration graph construction.
- (6) The access date in reference 30 should be updated.
- (7) The nominal content of dipentum capsule and colosalazine-EC should be added in Table 5.
- (8) The references corresponding to methods of assessment of greenness should be written in Table 6.
- (9) Please mention the reference methods details under validation of the proposed method
- (10) In the abstract: method permitted to methods permitted, evaluated against "green" criteria: regarding "green" criteria

Decision letter (RSOS-210110.R0)

Dear Dr Elmansi:

Title: Studying the quenching resulted from the formation of association complex between olsalazine or sulfasalazine with acriflavine
Manuscript ID: RSOS-210110

Thank you for submitting the above manuscript to Royal Society Open Science. On behalf of the Editors and the Royal Society of Chemistry, I am pleased to inform you that your manuscript will be accepted for publication in Royal Society Open Science subject to minor revision in accordance with the referee suggestions. Please find the reviewers' comments at the end of this email.

The reviewers and handling editors have recommended publication, but also suggest some minor revisions to your manuscript. Therefore, I invite you to respond to the comments and revise your manuscript.

Because the schedule for publication is very tight, it is a condition of publication that you submit the revised version of your manuscript before 10-Mar-2021. Please note that the revision deadline will expire at 00.00am on this date. If you do not think you will be able to meet this date please let me know immediately.

When submitting your revised manuscript, you will be able to respond to the comments made by the referees and upload a file "Response to Referees" in "Section 6 - File Upload". You can use this to document any changes you make to the original manuscript. In order to expedite the

processing of the revised manuscript, please be as specific as possible in your response to the referees.

Kind regards,
Dr Laura Smith
Publishing Editor, Journals

RSC Associate Editor:
Comments to the Author:
(There are no comments.)

RSC Subject Editor:
Comments to the Author:
(There are no comments.)

Reviewer comments to Author:

Reviewer: 1

Comments to the Author(s)

I highly recommend the manuscript for publication after minor revision as below.

Suggestion to Authors

1- Please modify the Abstract section as below.

We report the detection and quantification of important ulcerative colitis drugs olsalazine (OLS) and sulfasalazine (SUL) by the Spectrofluorometric method. The proposed method was optimized and validated by using the quenching effect on the acriflavine fluorescence. The method was applied on the detection and quantification of OLS and SUL under optimized conditions showing the calibration curves were linear (Range: 1.0–10.0 $\mu\text{g/mL}$), with correlation coefficients $R^2 = 0.9999$ for both the drugs. The limits of detection (LOD) and quantification (LOQ) were 53 ng/mL & 104 ng/mL for the OLS and 160 ng/mL & 315 ng/mL for the SUL. This method permitted the analysis of OLS and SUL in their pure and pharmaceutical forms. The proposed spectrofluorimetric method was also evaluated against “green” criteria and all the experimental results make it an eco-friendly and safe method for the detection of OLS and SUL.

2- Please modify the Conclusion section as below.

The proposed spectrofluorimetric method was effectively utilized to estimate OLS and SUL by measuring their quenching effect on the acriflavine fluorescence. The proposed method was optimized, validated and applied successfully for the pharmaceutical preparations of the OLS and SUL. The static quenching mechanism was confirmed by the decrease in Stern–Volmer constants with the increasing temperature. The method is facile, sensitive and economical alternative to previously reported analytical methods that consume expensive or hazardous solvents with complicated instrumentation. Additionally, the spectrofluorimetric method is “green” thereby recommended for safety, applicability and eco-friendly nature.

Reviewer: 2

Comments to the Author(s)

It may be published after a revision considering the following points:

- (1) The purity of olsalazine and sulphasalazine should be written under section 2.1
- (2) Under section 2.1 "OLS and SUL were kindly provided by sigma" should be changed to "OLS and SUL were purchased from Sigma"; in addition country should be added.
- (3) Under material, reagent and standard solution: "filling to volume" should be changed to "diluted to volume".
- (4) Effect of surfactant should be added under optimization of experiment section 3.1
- (5) The sensitivity of instrument should be added under section 2.3.1 calibration graph construction.
- (6) The access date in reference 30 should be updated.
- (7) The nominal content of dipentum capsule and colsalazine-EC should be added in Table 5.
- (8) The references corresponding to methods of assessment of greenness should be written in Table 6.
- (9) Please mention the reference methods details under validation of the proposed method
- (10) In the abstract: method permitted to methods permitted, evaluated against “green” criteria: regarding “green” criteria

Author's Response to Decision Letter for (RSOS-210110.R0)

See Appendix A.

Decision letter (RSOS-210110.R1)

Dear Dr Elmansi:

Title: Studying the quenching resulted from formation of an association complex between olsalazine or sulfasalazine with acriflavine
Manuscript ID: RSOS-210110.R1

It is a pleasure to accept your manuscript in its current form for publication in Royal Society Open Science. The chemistry content of Royal Society Open Science is published in collaboration with the Royal Society of Chemistry.

RSC Associate Editor
Comments to the Author:
(There are no comments.)

Reviewer(s)' Comments to Author:

Appendix A

Thank you for the editorial office of Royal Society Open Science for giving us opportunity to further revise our manuscript. All the valuable reviewer comments were respected and taken into consideration. The letter including our reply to the comments.

Studying the quenching resulted from formation of an association complex between olsalazine or sulfasalazine with acriflavine

Manuscript ID: RSOS-210110

Reviewer1:

1- Please modify the Abstract section as below. We report the detection and quantification of important ulcerative colitis drugs olsalazine (OLS) and sulfasalazine (SUL) by the Spectrofluorometric method. The proposed method was optimized and validated by using the quenching effect on the acriflavine fluorescence. The method was applied on the detection and quantification of OLS and SUL under optimized conditions showing the calibration curves were linear (Range: 1.0–10.0 µg/mL), with correlation coefficients $R^2 = 0.9999$ for both the drugs. The limits of detection (LOD) and quantification (LOQ) were 53 ng/mL & 104 ng/mL for the OLS and 160 ng/mL & 315 ng/mL for the SUL. This method permitted the analysis of OLS and SUL in their pure and pharmaceutical forms. The proposed spectrofluorimetric method was also evaluated against "green" criteria and all the experimental results make it an eco-friendly and safe method for the detection of OLS and SUL.

- Reply: The abstract has been modified as recommended by the reviewer.

2- Please modify the Conclusion section as below. The proposed spectrofluorimetric method was effectively utilized to estimate OLS and SUL by measuring their quenching effect on the acriflavine fluorescence. The proposed method was optimized, validated and applied successfully for the pharmaceutical preparations of the OLS and SUL. The static quenching mechanism was confirmed by the decrease in Stern–Volmer constants with the increasing temperature. The method is facile, sensitive and economical alternative to previously reported analytical methods that

consume expensive or hazardous solvents with complicated instrumentation. Additionally, the spectrofluorimetric method is "green" thereby recommended for safety, applicability and eco-friendly nature.

- Reply: The conclusion has been modified as recommended by the reviewer.

Reviewer2:

(1) The purity of olsalazine and sulphasalazine should be written under section 2.1

- Reply: The purities of olsalazine and sulphasalazine were written.

(2) Under section 2.1 "OLS and SUL were kindly provided by sigma" should be changed to "OLS and SUL were purchased from Sigma"; in addition country should be added.

- Reply: The phrase was corrected as recommended.

(3) Under material, reagent and standard solution: "filling to volume" should be changed to "diluted to volume".

- Reply: The phrase was changed as requested.

(4) Effect of surfactant should be added under optimization of experiment section 3.1

- Reply: Effect of surfactant was included accordingly

(5) The sensitivity of instrument should be added under section 2.3.1 calibration graph construction.

- Reply: sensitivity of the instrument was added as requested

(6) The access date in reference 30 should be updated.

- Reply: The access date was updated.

(7) The nominal content of dipentum capsule and colsalazine-EC should be added in Table 5.

- Reply: The nominal content were calculated and inserted in Table 5.

(8) The references corresponding to methods of assessment of greenness should be written in Table 6.

- Reply: Access date was updated.

(9) Please mention the reference methods details under validation of the proposed method

Reply: the reference methods details were added as requested.

(10) In the abstract: method permitted to methods permitted, evaluated against "green" criteria: regarding "green" criteria

- Reply: The abstract section was rephrased totally in accordance with reviewers comments.